# Trench Formation under the Tunable Nanogap: Its Depth Depends on Maximum Strain and Periodicity

**DOI:** 10.3390/mi14111991

**Published:** 2023-10-27

**Authors:** Daehwan Park, Dukhyung Lee, Mahsa Haddadi Moghaddam, Dai-Sik Kim

**Affiliations:** Department of Physics, Ulsan National Institute of Science and Technology (UNIST), Ulsan 44919, Republic of Korea; mossmouse@unist.ac.kr (D.P.); hyung0624@unist.ac.kr (D.L.); mahsahm@unist.ac.kr (M.H.M.)

**Keywords:** nanogap, zerogap, flexible substrate, crack, COMSOL, PDMS

## Abstract

Metallic nanogaps have been studied for many years in the context of a significant amount of field enhancements. Nanogaps of macroscopic lengths for long-wave applications have attracted much interest, and recently one dimensional tunable nanogaps have been demonstrated using flexible PET substrates. For nanogaps on flexible substrates with applied tensile strain, large stress is expected in the vicinity of the gap, and it has been confirmed that several hundred nanometer-deep trenches form beneath the position of the nanogap because of this stress singularity. Here, we studied trench formation under nanogap structures using COMSOL Multiphysics 6.1. We constructed a 2D nanogap unit cell, consisting of gold film with a crack on a PDMS substrate containing a trench beneath the crack. Then, we calculated the von Mises stress at the bottom of the trench for various depths and spatial periods. Based on it, we derived the dependence of the trench depth on the strain and periodicity for various yield strengths. It was revealed that as the maximum tensile strain increases, the trench deepens and then diverges. Moreover, longer periods lead to larger depths for the given maximum strain and larger gap widths. These results could be applied to roughly estimate achievable gap widths and trench depths for stretchable zerogap devices.

## 1. Introduction

In the context of light–matter interaction at a subwavelength scale, the narrow gap formed between two metal objects is of critical importance, because charge conservation demands that charges accumulate at the metal facets, driven by the surface currents induced by the magnetic vector of light [1]. Those charges are responsible for the electric field between the gap being much larger than the incident electric field of light. This field enhancement phenomenon inside a nanogap is achieved between metal nanoparticles, a metallic tip and a flat metal surface, or inside a linear trench, which are taken advantage of when studying nanogap-related physics or in surface-enhanced Raman scattering (SERS), tip-enhanced Raman scattering (TERS), plasmon-exciton coupling, nonlinear response of materials, sensing over diffraction limit, quantum conductance, etc. [2,3,4,5,6,7].

Among the various forms of nanogaps, an infinitely long slit has effective geometry for large electric field enhancement. In the early stage of studies on extraordinary optical transmission [8], it has been theoretically and experimentally demonstrated that shape resonance itself can cause the extraordinary transmission of light through an isolated subwavelength hole [9,10]. The transmission amplitude normalized by hole area is an indicator of the amount of the field enhancement within the hole, and it keeps increasing with the higher aspect ratios of the subwavelength hole. According to this tendency, an infinitely long slit structure with infinitesimal width is expected to achieve the highest field enhancement at the lowest frequency, and this scheme holds for slits narrower than the skin depth of metal [11]. The development of nanogap technology enables the manipulation of the magnetic field of light using its complementary structure based on Babinet’s principle [12,13]. Moreover, the development of atomic layer lithography techniques enables the photolithography-associated arbitrary design of nanogap devices, which improves scalability all the way to macroscopic wafers [14]. Thanks to the fabrication technique, the ultimate limit of the field enhancement in a classical regime was explored, and it was found that the field enhancement factor was saturated as gap size decreased to 1.5 nm and achieved 2000 at 0.15 THz [15]. Recently, Ångstrom gaps with van der Waals scale widths have been employed to study the optical properties of graphene [16].

Since the resonance properties of nanogap structures depend on structural parameters such as gap width, particularly the nature of the gap plasmon, it would be beneficial to develop nanogap devices with tunable geometric parameters. To develop such tunable nanogap devices, flexible substrates such as polyethylene terephthalate (PET) have been employed, and mechanically tunable twofold molecule sensors and topology-changing metamaterials were demonstrated [17,18]. In [18], alumina-filled nanogap rings manufactured by atomic layer lithography fabricated on a PET substrate, the gap was etched out to enable the closing of the gap by the bending of the substrate. The gap was closed all the way to contact with zero electrical resistance, which is critical to drastically change the topology and geometry of the nanogap ring structure. Another novel method was developed whereby the dielectric spacer was wholly absent, but a crack appeared with bending precisely along photolithographically defined lines. The gap width then repeatedly changed from zero to a few hundred nanometers, enabling the modulation of the transmitted microwave with a performance of 50 dB [19]. The addition of a metal spacer layer between the first and second deposition layers allowed cracks along the spacer line to be more repeatable [20]. We refer to this spacerless tunable gap technology as *zerogap technology*. For theoretical study using structural simulation, Ångstrom scale control of nanogaps by means of mechanical strain was explored [21].

When outer-bending is applied to PET zerogap devices, significant deformation occurs on the exposed surface of the substrate not covered by metal film, i.e., along the lithographically defined lines. This deformation induces substantial stress, which can lead to surface damage and the formation of trenches below the nanogap. The presence of these trenches beneath the nanogap has been reported in several studies [22,23]. It is noteworthy that these trenches are not mere degradation; they serve as an effective method to enhance the device’s robustness by relieving the strain singularity. Although early studies were primarily focused on preventing cracks on metal film, [24,25], recent studies have shifted their focus to the generation of these cracks in pre-designed positions [26,27]. Through the fabrication of micro-notches on the substrate, stress concentration is induced at these notches [26]. When the notches are designed on the top film, external strain can initiate crack propagation within the substrate along the path connecting those notches, leading to the formation of sub-micrometer-depth trenches; a 50 μm width film created by this crack development showed no random cracks even at 35% tensile strain [27]. Recently, trenches below zerogaps were observed using AFM scanning and optical diffraction [23].

As another form of zerogap technology, larger strain can be applied, and larger gaps expected when using polydimethylsiloxane (PDMS). This approach offers distinct advantages over the bending method. Although substrate thinning due to the Poisson effect is tricky for optical analysis, the gap width can be tuned with negligible displacement. Additionally, the inherent periodicity of stretchable devices leads to spatial uniformity in transmission properties. For stretchable PDMS electronic devices, it has been demonstrated that trenches could be developed due to the emergence of cracks on the film [27]. Hence, it would be valuable to study stretchable zerogap devices considering the presence of trenches with some predictive powers.

The purpose of this study was to investigate the mechanical properties of the suggested nanogap devices for application on PDMS. We presented a finite element analysis (FEA) based on the computational approach to this topic. Von Mises stress was utilized to predict the stability of a metallic structure involving trenches. First, we presented von Mises stress at the trench bottom with various trench depths and tensile strains. We indirectly studied the rupture process of the suggested devices, and finally suggested the dependence of the allowed gap widths’ on two design factors: the yield strength of PDMS and the spatial period of the device. It is anticipated that the trench and gap widths of a few nanometers to a few microns are possible with modest strain.

## 2. Methods

We employed the solid mechanics module of COMSOL Multiphysics 6.1 to conduct mechanical simulations on the stretchable nanogap device. For the material properties of both gold and PDMS, we utilized the pre-defined parameters available in the MEMS section of the COMSOL material library. These parameters were as follows: the density, Young’s modulus, and Poisson ratio were 19,300 kg/m^3^, 70 GPa, and 0.44 for gold, and 970 kg/m^3^, 750 kPa, and 0.49 for PDMS. Since both materials possess a high Poisson ratio, they are regarded as nearly incompressible materials. To describe elasticity, we treated gold as an isotropic linear elastic material. In the case of the PDMS substrate, we applied the hyperelastic material model [28]. It is important to note that the elastic properties of PDMS depend on the mixing ratio between the base polymer and the curing agent, resulting in a diverse range yield strength. In our simulations, we adopted parameters obtained from a mixing ratio of 15:1, which exhibit the highest stretchability among available data. The second order Odgen model was implemented, as it shows good agreement with the experimental data up to 150% of tensile strain compared with the Neo–Hookean or Mooney–Rivlin models [28]. The details are presented in Table 1.

We construct half of the nanotrench unit cell under the plane strain condition and used the mirror 2D dataset for visualization. Details are presented in Figure 1. In the nanotrench configuration, gold film is attached to a PDMS substrate, where their thicknesses are 100 nm and 200 μm, respectively. The optical properties of the thin gold film have marginal differences compared with thicker films [29]. A two hundred micron PDMS membrane can be obtained by conventional spin coating [30]. Five nanometer wide trenches are placed near the edge of the half unit cell, resulting in a 10 nm wide initial gap between neighboring gold films, and the bottom of the trench possesses a semicircle shaped cross section. The spatial period and trench depth range from 5 μm to 100 μm, and from 0.1 μm to 20 μm, respectively. The length of the trenches is set at 2 cm.

Gap width is measured at the top points of the adjacent metal films on either side of the crack. The yield strength of PDMS is assumed to be between 1 MPa and 10 MPa, taking into consideration the differing preparation methods such as curing conditions and mixing ratios [31,32]. For each trench depth, the tensile strain where the von Mises stress reaches the yield strength was obtained through linear interpolation. At a given strain level, the corresponding gap width was obtained through the same process.

## 3. Results

### 3.1. Tensile Strain Induced Trench Development

Figure 2 illustrates a 2D von Mises stress map of the cross section of the stretchable nanogap at various depths while applying the tensile strain. The initial trench depths were, respectively, (a) 0.1 μm, (b) 0.2 μm, (c) 0.5 μm, (d) 1 μm, (e) 2 μm, (f) 5 μm, and (g) 10 μm. The bottom of the trench (notch) shows the maximum von Mises stress. In the region far below the notch, a nearly uniform stress distribution was observed coalescing with the PDMS stress without nanostructures on top. Moving away from the crack towards the top gold film, a general trend of decreasing stress was found. Interestingly, we identified a cycloidal-shaped low-stress *blue-colored* path connecting the neighboring trench bottoms for the two deepest trenches, Figure 2e,f [33]. At the middle of the gold film, a low-stress *green-colored* horizontal line appeared. All figures represent critical situations in which the von Mises stress at the bottom, being always a maximum, was 8 MPa, assumed to be the yield strength. This yield criterion of PDMS falls within the reported range [32]. A trend was noticed: the deeper the trench, the larger the maximum tensile strain and the larger the gap width. However, for trenches deeper than 5 μm, the maximum tensile strain was saturated; in other words, a deeper trench does not result in higher overall strain at which the yield strength is achieved. This could imply rapid crack growth into the substrate, which is suggested in previous reports on crack extension in plain gold film on PDMS substrates [34,35]. It is noteworthy that a cycloid-like, low-stress line above which stress is much lower (overall blue colors) appears for deep trenches in which the tensile strain becomes saturated. This implies that the crack relieves much of the stress at the upper part of the structure. The height of cycloid-like path suffers few changes relative to the trench bottom while the trench depth varied. The vertical distance from the top of the low-stress path to the trench bottom shows a dependence on the period of the device, which is described in the discussion section.

Figure 3 presents a systematic analysis on the relationship between three parameters: trench depth, maximum tensile strain, and the period of the nanogap. Figure 3a describes the relations between trench depth and maximum tensile strain for 5 μm, 10 μm, and 20 μm of periodicity. In all cases, as described in Figure 2, there were two regimes—an initial logarithmic increase of the maximum tensile strain with respect to the trench depth followed by saturation. As expected, increasing the yield strength from 1 MPa to 10 MPa resulted in a higher maximum tensile strength. Interestingly, it appears that these two regimes are distinguished by the appearance of the cycloidal-like low-stress path for larger depths: saturation strains. A larger period resulted in a smaller saturation strain, implying that larger stress was applied to the gap region. Here, we defined the *critical tensile strain* to which the maximum tensile strain saturates in Figure 3a. Following this definition, Figure 3b and the inset, we show the dependence of the critical tensile strain on the yield strength of the PDMS and the period. The critical tensile strain decreases with a longer nano trench period, while the higher yield strength of PDMS results in increased critical tensile strain.

### 3.2. Effect of Maximum Tensile Strain on the Achievable Gap Width

In Figure 4, we estimate the maximum gap width that can be achieved for the given yield strength of the PDMS and the periodicity, for the application of stretchable zerogap devices. The maximum gap width was found at the critical tensile strain determined by these two parameters. For the nanogap device with a 5 μm period, the maximum gap widths across various yield strengths were less than a micrometer. The maximum gap width became larger as the period increased: a 100 μm period allowed a maximum gap width that ranged from 1 μm to 4 μm. Higher yield strength allowed wider maximum gap widths. For instance, when the yield strength of PDMS increased 10 times, from 1 MPa to 10 MPa, the maximum gap width increased about 4 times. These results suggest that the maximum gap width of stretchable zerogap devices can be varied systematically based on the mechanical properties of the PDMS and pre-defined periodicity, as well as the depth of the periodic trenches.

In the systematic simulation of stretchable nanogap devices, the critical tensile strain and the maximum gap width are important quantities, and we have already shown that they depend on two control parameters: the period of the device and the yield strength of the PDMS. Those dependencies are summarized in Figure 5.

## 4. Discussion

When applying tensile strain to stretchable nanogap devices, the width of the nanogap can be expected to change by a few orders of magnitude: for example, assuming the yield strength of PDMS to be 10 MPa, a 100 μm period nanogap can be expanded from the 10 nm of the initial width to 3.6 μm. In fact, the 10 nm initial width is somewhat arbitrary and can be of sub-nanometer range as in zerogap technologies [19,20,23], making the range even larger. Due to the relatively high elastic modulus of gold compared to the substrate, most of the stretching occurred at the PDMS under the nanogap region. As a result, a trend was found in our calculations that the width of the nanogap is proportional to the tensile strain, as represented in Figure 2:(1)Gap width≅Tensile strain×nanogap period
and confirmed by experiments [23]. Obviously, to stretch the gap region by orders of magnitudes, damage is expected to occur because for bulk PDMS, it is hard to endure more than 200% of the tensile strain. This is why we expect the creation of trenches and eventual stress-relief below the gap, even without the intentional pre-patterning of the trenches. In our work, we assume the existence of trenches and use the yield strength to backtrack the trench depth created by PDMS stretching. Due to the presence of these trenches, the device could endure stress until the von Mises stress at the bottom of the trench equals the yield strength of the substrate; otherwise, cracks will propagate through the substrate. In that sense, we *indirectly* investigated the development of cracks and their depth in our stretchable nanogap device.

We explain the existence of two low-stress paths in the 2D von Mises stress map displayed in Figure 2. For the *blue-colored* path in the PDMS substrate, which appears in Figure 2f,g, we might consider the relationship between the cycloid cusp structure and the periodic trench structure [33]. At the cycloidal path in the periodic crack structure, substantially less stress is expected compared to the surroundings. The stress distribution within the gold film could be understood through the concept of the neutral plane [36]. When we subject the trenched device to tensile strain in the presence of the gold film on top, it begins to curve inward, resulting in the formation of a neutral plane within the film at a specific height that satisfies Equation (2):(2)$$Y_{Top}\times {d_{Top}}^{2}=Y_{Bottom}\times {d_{Bottom}}^{2}$$
where *Y* and *d* represent Young’s modulus and the distance from the surface, respectively. Since we are dealing with a uniform gold film, a *green-colored* stress-free area should appear in the middle of the film, as described in Figure 2.

Now, we will discuss fabrication implications. To investigate the role of trenches experimentally in stretchable zerogap devices, several methods have been reported for developing and detecting trenches; applying a UV treatment on fabricated PDMS devices induces hardening, which can facilitate cracking through pre-defined notches [27]. For profile measurement, fabricating the mold of trenched PDMS followed by silane treatment could be utilized [35]. Note that the actual device could give somewhat different results from this simulation, since the viscoelastic behavior of PDMS is not regarded in this simulation and nonlinear effects are only guessed. [37] The mechanical properties of gold film for zerogaps are also important for preventing uncontrolled cracks on the gold film itself. When fabricating long-period stretchable zerogap devices for long-wave applications, the mechanical properties of gold films on zerogaps resemble those of plain gold film, whose tensile strain maximum is no more than 1% [38]. It could be expected that gold film on zerogaps is more vulnerable as the period increases, leading to a reduction in the achievable maximum gap width.

## 5. Conclusions

We studied the role of periodic trenches formed beneath nanogaps in stretchable zerogap devices and attempted to estimate the achievable gap width based on 2D finite element analysis. We calculated the von Mises stress in the provided structure while varying the periods of the nanotrenches and the yield strength of the PDMS. Based on these results, for several of the set trench depths, we found the critical tensile strain level by linear interpolation and by assuming a yield strength value. We found the relationship between trench depth and tensile strain and estimated the maximum gap width. Features in 2D stress maps were analyzed in terms of neutral planes and stress singularities on periodic surfaces. This work could be utilized as design principles for highly stretchable crack-based electronic and optical devices.

## Figures and Tables

**Figure 1 micromachines-14-01991-f001:**
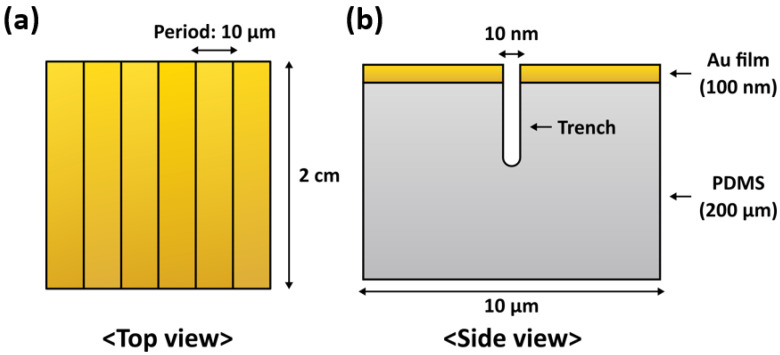
Schematics of our stretchable nanogap device with 10 μm period. (**a**) Top view. (**b**) Side view.

**Figure 2 micromachines-14-01991-f002:**
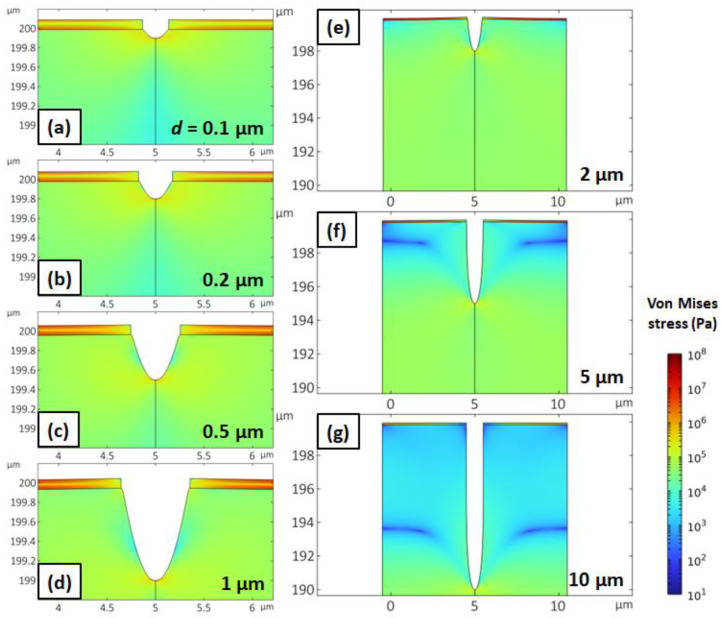
Cross-section of the trench in a stretchable nanogap with a 10 μm period under various applied tensile strains and initial trench depths, keeping the strain at the bottom of the trench constant at 8 MPa, simulating the high end of the yield-rupture strain. Trench depths are (**a**) 0.1 μm, (**b**) 0.2 μm, (**c**) 0.5 μm, (**d**) 1 μm, (**e**) 2 μm, (**f**) 5 μm, and (**g**) 10 μm. For each trench depth, the cross-section shape is obtained at tensile strains of 2.6%, 3.4%, 5%, 7%, 9%, 10%, and 10%, respectively.

**Figure 3 micromachines-14-01991-f003:**
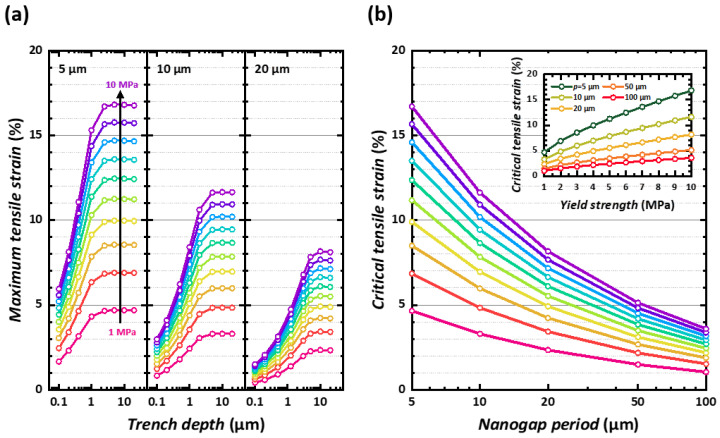
Maximum tensile strain for the given PDMS yield strength and device period. (**a**) Trench depth versus maximum tensile strain for 1 to 10 MPa of the yield strength; 5 μm, 10 μm, and 20 μm periods are simulated. (**b**) Estimated maximum tensile strain versus device period, for 1 to 10 MPa of PDMS yield strength and for 5 μm to 100 μm periods. Inset: maximum tensile strain versus the yield strength.

**Figure 4 micromachines-14-01991-f004:**
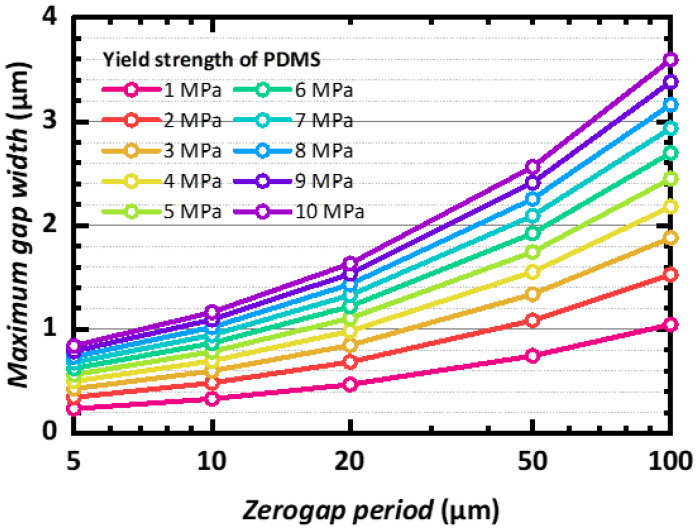
Maximum gap width of the stretchable zerogap for given device periods of 5 μm to 100 μm and PDMS yield strengths of 1 MPa to 10 MPa.

**Figure 5 micromachines-14-01991-f005:**
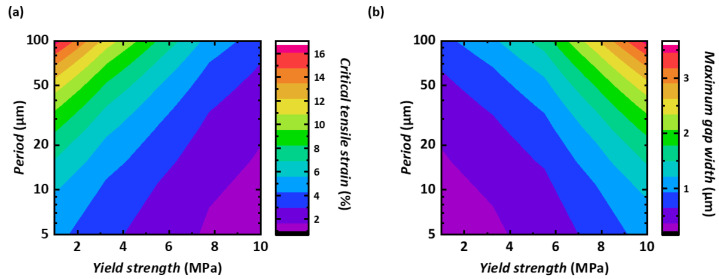
2D contour mapping of (**a**) critical tensile strain and (**b**) maximum gap width, in terms of the period and the yield strength of the PDMS.

**Table 1 micromachines-14-01991-t001:** Hyperelastic parameters for PDMS described using the two-term Odgen model.

μ_1_	μ_2_	α_1_	α_2_
0.244339 MPa	0.0146323 MPa	1.01795	3.74094

## Data Availability

The data presented in this study are available on request from the corresponding author.

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
