# Peer review of "Trench Formation under the Tunable Nanogap: Its Depth Depends on Maximum Strain and Periodicity"

_micromachines, 2023, doi:10.3390/mi14111991_

Round 1

Reviewer 1 Report

Comments and Suggestions for Authors

The authors revealed the relationship between von Mises stress, depth, and spatial period of a bottom trench on PDMS by employing COMSOL. The manuscript is well organized the conclusion is supported by the data. I recommend it publication after minor modification.

1.    The language needs to be polished., e.g., the sentence “The nanogap of macroscopic lengths for long-wave applications have attracted much interest and recently, one dimensional tunable nanogap has been demonstrated using flexible PET substrate.” can be changed to “The nanogap of macroscopic lengths for long-wave applications have attracted much interest, and recently one dimensional tunable nanogap has been demonstrated using flexible PET substrate.”. The sentence “we derive dependence of the trench depth to strain and periodicity for various yield strengths” can be changed to “we derive dependence of the trench depth on strain and periodicity for various yield strengths”. A full stop is missed in the sentence ‘’The gap width then changes from zero to a few hundred nanometers repeatedly, to enable the modulation of the transmitted 66 microwave with a 50 dB performance [18] An addition of a metal spacer layer between the first and second deposition layers allows crack along the spacer line more repeatable.‘’

2.    The square frame around the (a), (b), (c) in Figure 1 can be deleted.

3.    It is a little bit difficult to fully understand the Figure 5, or it seems not important.

4.    This study is to investigate the mechanical properties of the suggested stretchable nanogap device for application on bending PDMS. While the horizontal stretching force applied on PDMS is not investigated. May be the author can further investigate the dynamical response of the PDMS when an external mechanical stretching in horizontal direction is applied on the PDMS (refer to Small Methods 2023, 2201427; Appl. Phys. Lett. 2003, 82, 344 2404 published by different groups) in the future, or give a discussion in this manuscript

5.    I am wonder if it is appropriate to call the “tunable nanogap” as “zerogap”, even though the gap size can be changed from zero to hundreds of nanometers.

6.    May it is better to high light the novelty of this study compared to the authors’ previous publication (Strain versus tunable terahertz nanogap width: A simple formula and a trench below. Nanomaterials 2023, 13, 2526).

Comments on the Quality of English Language

Need to be checked.

Reviewer 2 Report

Comments and Suggestions for Authors

The paper deals with a topic, the so-called “zerogap” devices, already widely treated by the same group of Authors in several recent publications. Roughly speaking, a gold film is deposited over a deformable substrate and patterned by lithography: application of a longitudinal strain to the system leads to locally fracture the film, thus creating nano- or micro-sized gaps, with potential applications in plasmonics and related fields.

Compared to other publications on the same topic, the present paper shows quite a limited scope, being focused on numerical simulation of mechanical effects, in particular the stress distribution within the substrate is found as a function of several hypothetical process parameters. There is no evidence of original aspects in the mechanical model, and a commercial software is used.

Nonetheless, the paper can be considered for publication, since, further to the specific application to “zerogap” devices, simulating the stress distribution in microscale can be of some interest also for other systems or applications. Moreover, contrary to previous implementations involving, e.g., PET, PDMS, an interesting soft matter substrate, is employed. 

Prior to acceptance, Authors are requested to account for the following criticisms and prepare a revised version of the manuscript, accordingly.

1.     The hyperelastic model is mentioned at page 3, but no explanation is provided, nor motivations why it eventually leads to an improved description, with respect to conventional elastic models, are outlined. Authors must add some lines of discussion in the text.

2.     It is clear that the paper does not involve practical aspects dealing with feasibility of the proposed structures, however some discussion should appear in the text, for instance highlighting possible methods to practically realize the 10 nm trenching envisioned in the paper (see, e.g., Fig. 1 and relevant description). I think the only possibility would be electron lithography: if I well understood, the system would comprise of 10 nm trenches with a rather large aspect ratio (I see a trench depth in the micrometer range). Authors should add a discussion in the text addressing practical possibilities to obtain such large aspect ratio structures in real devices.

3.     Another point deserving discussion in the text deals with the PDMS substrate fabrication. Authors mention spin coating: is it feasible to have (good quality, i.e., homogeneous) 2 micrometer thick PDMS films by using spin coating? A brief discussion, along with relevant references, should be added in the text.

4.     Also the deposited gold film may be an issue in practical devices, certainly worth of a discussion. Adhesion of the metal onto the PDMS substrate is not considered at all in the paper: do Authors believe adhesion is not playing any role? Often, a thin intermediate layer of, e.g., chromium is used to improve adhesion: would it modify the presented results?

5.     A “neutral plane” is mentioned at line 248 and following: Authors must better explain what they intend for “neutral plane”.

6.     I deem Fig. 5 almost useless: Authors may consider removing it from the paper.

7.     There are a few syntax/style issues such as, for instance:

a)     Line 32: “are taken advantage”

b)    Line 50: “and find that”

c)     Line 55:  what does it mean “the gap plasmon nature”?

d)    Line 61: “The gap closes all the way to contact; zero electrical resistance…” (punctuation?)

e)    Line 181-2: “larger depths: saturation strains.”

f)      Line 215: “we already show…” (maybe we have already shown)

Comments on the Quality of English Language

Several sentences are unclear, in particular those listed in my comments. 

Reviewer 3 Report

Comments and Suggestions for Authors

Authors have attempted to study the structural behavior of a metallic nanogap on a flexible substrate when subjected to tensile strain. It explores the formation of trenches beneath the nanogap due to the stress generated by the strain. The investigation uses finite element analysis and aims to estimate the achievable gap width for stretchable "zerogap" devices.

The language was fairly easy to understand. 

Authors have performed several investigations, which is quite commendable. 

Understanding these aspects of your study can provide valuable insights into the potential applications and implications of your findings in the field of stretchable electronic and optical devices.
